# Effects of plasminogen activator inhibitor-1 deficiency on bone disorders and sarcopenia caused by adenine-induced renal dysfunction in mice

**Yuya Mizukami, Naoyuki Kawao, Takashi Ohira, Kiyotaka Okada, Hisatoshi Yamao, Osamu Matsuo, Hiroshi Kaji** *

Kindai University Faculty of Medicine, Department of Physiology and Regenerative Medicine, Osakasayama, Osaka, Japan

* hkaji@med.kindai.ac.jp

**Data Availability Statement:** All relevant data are within the paper and its Supporting Information files.

## Abstract

Chronic kidney disease (CKD) is a significant global health issue and often involves CKD-mineral and bone disorder (MBD) and sarcopenia. Plasminogen activator inhibitor-1 (PAI-1) is an inhibitor of fibrinolysis. PAI-1 has been implicated in the pathogenesis of osteoporosis and muscle wasting induced by inflammatory conditions. However, the roles of PAI-1 in CKD-MBD and sarcopenia remain unknown. Therefore, the present study investigated the roles of PAI-1 in bone loss and muscle wasting induced by adenine in PAI-1-deficient mice. CKD was induced in PAI-1$^{+/+}$ and PAI-1$^{-/-}$ mice by administration of adenine for ten weeks. Muscle wasting was assessed by grip strength test, quantitative computed tomography (CT) analysis and muscle weight measurement. Osteoporosis was assessed by micro-CT analysis of femoral microstructural parameters. PAI-1 deficiency did not affect adenine-induced decreases in body weight and food intake or renal dysfunction in male or female mice. PAI-1 deficiency also did not affect adenine-induced decreases in grip strength, muscle mass in the lower limbs, or the tissue weights of the gastrocnemius, soleus, and tibialis anterior muscles in male or female mice. PAI-1 deficiency aggravated trabecular bone loss in CKD-induced male mice, but significantly increased trabecular bone in CKD-induced female mice. On the other hand, PAI-1 deficiency did not affect cortical bone loss in CKD-induced mice. In conclusion, PAI-1 is not critical for the pathophysiology of CKD-MBD or CKD-induced sarcopenia in mice. However, PAI-1 may be partly related to bone metabolism in trabecular bone in the CKD state with sex differences.

## Introduction

Chronic kidney disease (CKD) represents a significant global health issue with substantial morbidity and mortality [1]. Approximately 700 million individuals worldwide are currently estimated to have CKD [2], and the number of CKD patients has continued to increase.

**Funding:** This work was supported by the following grants: a JSPS KAKENHI Grant-in-Aid for Early Career Scientists (No. 22K16755) to Y.M. and Grants-in-Aid for Scientific Research (No. C:20K09514; No. C:KK230021) to H.K. (https://www.jsps.go.jp/j-grantsinaid/); The Salt Science Research Foundation (No. 22C1) to H.K. (https://www.saltscience.or.jp/research/). No funders play role in the study design, data collection and analysis, decision to publish, or preparation of the manuscript.

**Competing interests:** The authors have declared that no competing interests exist.

According to the Global Burden of Disease study, the global prevalence of CKD increased by 33% between 1990 and 2017 [2]. CKD patients are frequently complicated with CKD-mineral and bone disorder (CKD-MBD), which is a complex condition involving mineral disturbances and bone disorders associated with abnormal vascular calcification and endocrine dysregulation [3]. In the pathophysiology of CKD-MBD, the progression of renal dysfunction induces vitamin D deficiency, hyperphosphatemia related to elevated fibroblast growth factor 23 (FGF23), excessive oxidative stress, and hyperparathyroidism [3]. Moreover, sarcopenia is a frequent finding in severe CKD [4]. The progression of CKD-MBD and sarcopenia has a negative impact on quality of life and worsens skeletal disorders, such as bone fractures. Consequently, CKD is one of the escalating global health concerns in need of resolution. Although we previously reported that irisin was partly involved in osteopenia induced by renal insufficiency as a myokine in the linkage of muscle to bone in mice, the pathogenesis of CKD-MBD remains unclear [5].

Plasminogen activator inhibitor 1 (PAI-1) is a member of the serine protease inhibitor superfamily and serves as a primary inhibitor of the fibrinolytic system under physiological and pathophysiological conditions [6]. Moreover, accumulating evidence indicates that PAI-1 plays a role in the regulation of bone remodeling and muscle wasting [7]. We previously demonstrated that PAI-1 deficiency prevented bone loss and delayed bone repair induced by diabetes and glucocorticoid excess in mice [8–10] and observed sex differences in the involvement of PAI-1 deficiency in osteopenia induced by a diabetic state in mice [11]. We also reported that PAI-1 deficiency inhibited the early differentiation of mouse mesenchymal stem cells to osteoblastic cells [12]. Moreover, we showed that endogenous and exogenous PAI-1 both decreased osteoblast activity in female mice, but not male mice [11,13]. In muscle tissues, we found that PAI-1 deficiency attenuated grip strength impaired by the diabetic state in female mice [14]. Collectively, these findings suggest the involvement of PAI-1 in bone metabolism and muscle wasting under endocrine disorders and inflammatory conditions in a partly sex-specific manner.

Plasma PAI-1 levels are elevated in several chronic inflammatory states, including CKD [15], and the findings of previous studies using animal models support PAI-1 being a fibrosis-promoting molecule in the kidneys [16–18]. However, the roles of PAI-1 in the effects of renal dysfunction on muscle and bone have yet to be elucidated. Therefore, we hypothesized the involvement of PAI-1 in the pathogenesis of bone disorders and sarcopenia induced by CKD. The administration of adenine induces renal dysfunction in mice due to the mineralization of adenine metabolites within kidney tissues [19]. In the present study, we investigated the effects of PAI-1 deficiency on bone disorders and sarcopenia caused by renal dysfunction induced by the continuous administration of adenine in both male and female mice.

## Materials and methods

### Animals and ethical statement

PAI-1$^{+/+}$ and PAI-1$^{-/-}$ mice with a mixed C57BL/6J (81.25%) and 129/SvJ (18.75%) background were originally generated by Professor D. Collen at the University of Leuven, Belgium [20]. These mice were kindly provided by Professor D. Collen in 1996 and have been subsequently bred in the animal facility at Kindai University. To minimize the effects of the mixed mouse strain, we obtained male and female mice with heterozygous PAI-1 (PAI-1$^{+/-}$) gene deficiency by crossbreeding PAI-1$^{+/+}$ and PAI-1$^{-/-}$ mice. These heterozygous littermates were then repeatedly bred. For the present study, PAI-1$^{+/+}$ and PAI-1$^{-/-}$ mice were prepared by breeding homozygous littermates obtained from heterozygous breeding. The genotypes were determined by PCR analysis (S1 Fig). Consequently, the genetic background of the PAI-1$^{+/+}$

and PAI-1$^{-/-}$ mice used in this study is nearly identical. All mouse experiments were performed according to the Guide for the Care and Use of Laboratory Animals from the National Institutes of Health and the institutional guidelines for the use and care of laboratory animals at Kindai University. The protocol was approved by the Experimental Animal Welfare Committee of Kinki University (permit number: KAME-2022-073). The collection of computed tomography (CT) images was performed under 2% isoflurane. At the end of the experiment, all mice were sacrificed using an overdose of isoflurane. All efforts were made to minimize suffering.

## CKD model induced by the administration of adenine

CKD was induced by the continuous administration of adenine for 10 weeks, as previously described with some modifications [19]. PAI-1$^{+/+}$ and PAI-1$^{-/-}$ mice of both sexes were fed CE-2 diets until they reached 12 weeks of age. Both sexes were then divided into four groups: PAI-1$^{+/+}$/Control (n = 8), PAI-1$^{+/+}$/Adenine (n = 8), PAI-1$^{-/-}$/Control (n = 8), and PAI-1$^{-/-}$/Adenine (n = 8). To induce renal dysfunction, 12-week-old mice were fed CE-2 diets containing 0.25% adenine for 2 weeks. The diets were then switched to CE-2 diets containing 0.15% adenine for the next 8 weeks. Control groups were fed CE-2 diets containing no adenine for the entire 10-week period. Food intake and body weight were measured twice a week. Ten weeks after the initiation of adenine administration, mice underwent grip strength tests and were scanned using an X-ray CT system (Latheta LCT-200; Hitachi Aloka Medical, Tokyo, Japan) or a μCT system (Cosmo Scan GX II, Rigaku Corporation, Tokyo, Japan). Following the collection of blood samples under 2% isoflurane anesthesia, mice were euthanized with excess isoflurane. The gastrocnemius (GA), soleus, and tibialis anterior (TA) muscles were isolated, and their wet weights were measured.

## Blood chemistry

Serum levels of blood urea nitrogen (BUN), creatinine, calcium, phosphorus, and parathyroid hormone (PTH) were measured using the DetectX Urea Nitrogen Colorimetric Detection Kit (Arbor Assays, MI, USA), LabAssay Creatinine (Wako Pure Chemicals, Osaka, Japan), Calcium E-Test Wako (Wako Pure Chemicals), Phospha C-Test Wako (Wako Pure Chemicals), and enzyme-linked immunosorbent assay kits for mouse PTH (RayBiotech, Norcross, GA, USA, Cat. No. EIAM-PTH-1), respectively, in accordance with the manufacturers' instructions.

## Grip strength test

Grip strength was measured using a grip strength meter (1027SM, Columbus Instruments, Columbus, OH, USA) ten weeks after the initiation of adenine administration.

## Quantitative CT (qCT) analysis

Muscle mass in the whole body and muscle mass in the lower limbs were measured by using qCT system. In the qCT analysis of muscle volume, mice were scanned and analyzed using a Latheta LCT-200 experimental animal CT system. CT images were acquired with a voxel size of 96 × 192 × 1008 μm and the region of interest was defined as the whole body for analyses of muscle mass in the whole body (lean body mass). CT images were acquired with a voxel size of 48 × 48 × 192 μm and the region of interest was defined as the segment from the proximal end to the distal end of the tibia for analyses of muscle mass in the lower limbs. Muscle mass was

calculated by the formula, Muscle mass (g) = Muscle volume (cm3) × Muscle density (1.06 g/cm3), using LaTheta software (version 3.41).

## Micro-CT (μCT) analysis

In the μCT analysis of bone morphology, mice were scanned using a Cosmo Scan GX II μCT system. The following parameters used for μCT scans: tube voltage of 90 kV, tube current of 88 μA, and isotropic voxel size of $10 \times 10 \times 10$ μm. Prior to the analysis of the bone microstructure, raw images were reconstructed using CosmoScan GX ImageAnalysis Software (Rigaku Corporation) with an isotropic voxel size of 6.5 μm. Microstructural parameters of femurs were assessed using the visualization and analysis software, Analyze 14.0 (AnalyzeDirect, Inc., KS, USA). In the trabecular bone analysis, a 1-mm-thick region from the end of the growth plate was used, and the following parameters were assessed: trabecular bone mineral density (BMD), the bone volume fraction (BV/TV, the ratio of the segmented bone volume to the total volume of the region of interest), trabecular thickness (Tb.Th), trabecular number (Tb.N), and trabecular separation (Tb.Sp). In the cortical bone analysis, a 1-mm-thick region of the mid-diaphysis of the femur was used, and the following parameters were assessed: cortical tissue mineral density (CtTMD), cortical bone area (Ct.Ar), average cortical thickness (Ct.Th), cortical porosity (Ct.Po), total cross-sectional area inside the periosteal envelope (Tt.Ar), and the cortical area fraction (Ct.Ar/Tt.Ar).

## Statistical analysis

All data are expressed as the mean ± the standard error of the mean (SEM). All values used to create the graphs are summarized in S1 File. The significance of differences was evaluated using a one-way ANOVA followed by the Tukey–Kramer post hoc test. A simple regression analysis was performed using Spearman's rank nonparametric correlation test. The significance level was set to $p < 0.05$. All statistical analyses were performed using GraphPad PRISM 9 software (La Jolla, CA).

## Results

### Effects of PAI-1 deficiency on kidney disorders induced by administering adenine diets

CKD was induced in PAI-1$^{+/+}$ and PAI-1$^{-/-}$ mice of both sexes by administering adenine diets, as summarized in Fig 1A. The administration of adenine resulted in significantly lower body weights and food intake in both PAI-1$^{+/+}$ and PAI-1$^{-/-}$ mice than in the control group (Fig 1B and 1C). Serum levels of BUN and creatinine were elevated after the administration of adenine, with similar levels being observed in PAI-1$^{+/+}$ and PAI-1$^{-/-}$ mice (Fig 1D and 1E). The administration of adenine increased the serum levels of calcium and phosphorus in male PAI-1$^{+/+}$ and PAI-1$^{-/-}$ mice. However, in female mice, serum calcium levels were significantly elevated in PAI-1$^{-/-}$ mice only, while serum phosphorus levels were significantly elevated in PAI-1$^{+/+}$ mice only (Fig 1F and 1G). The effects of adenine administration on serum PTH levels were not significant in all groups, although adenine administration tended to elevate serum PTH levels without any significant differences in female mice and PAI-1 deficiency significantly increased serum PTH levels in control male mice (Fig 1H). Serum PTH levels were significantly and positively corelated to serum calcium levels in only female mice, but not male, in the simple regression analyses (S2 Fig).

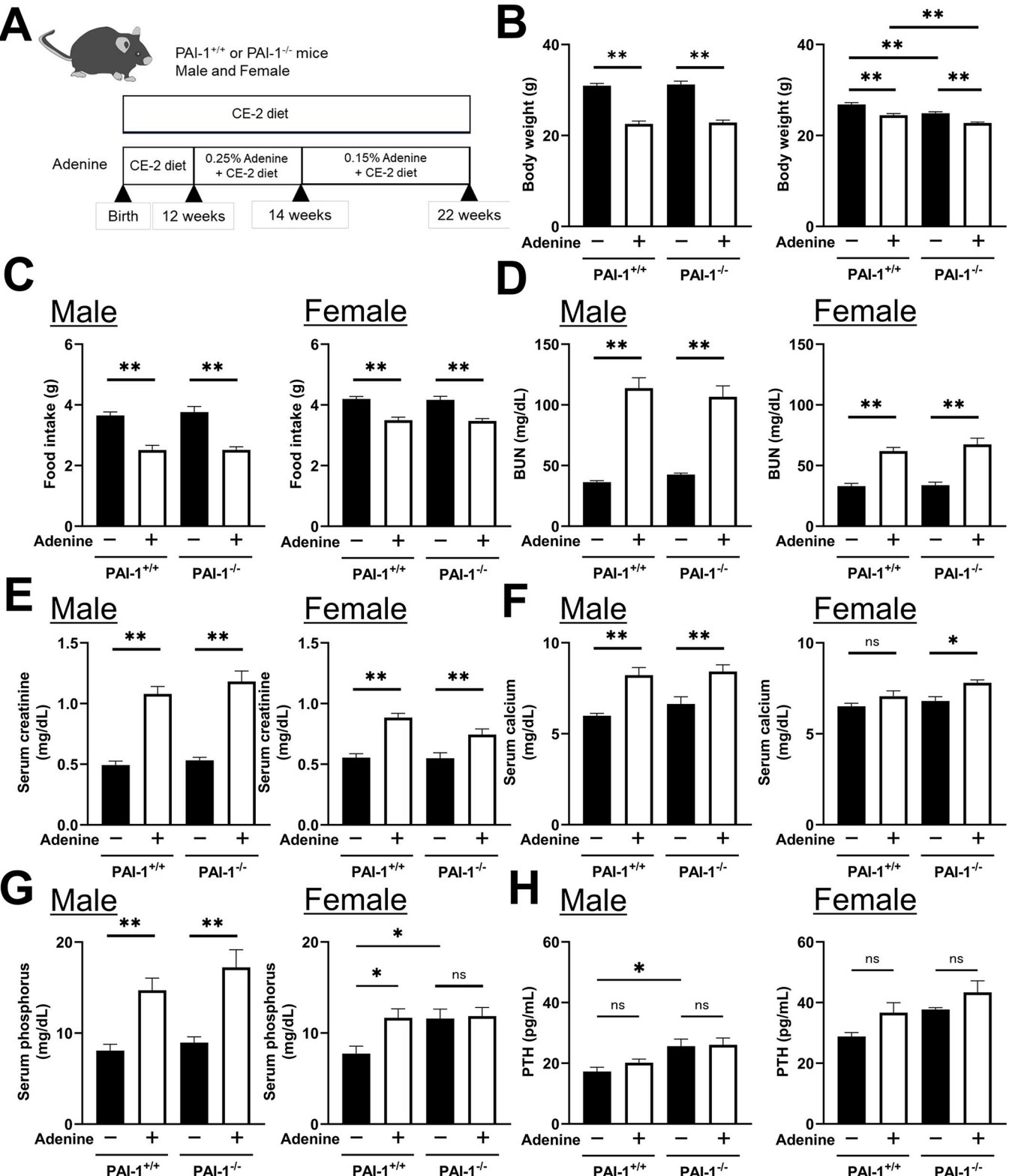

**Fig 1. Effects of PAI-1 deficiency on kidney disorders induced by administering adenine diets.** (A) A flow chart for the preparation of CKD model mice. (B-G) Body weight (B) food intake (C), serum blood urea nitrogen (BUN) levels (D), serum creatinine levels (E), serum calcium levels (F), serum phosphorus levels (G) and serum PTH levels in PAI-1$^{+/+}$ and PAI-1$^{-/-}$ mice ten weeks after the initiation of adenine administration. Results are expressed as the means ± SEM of 8 mice per group. Statistical analyses were performed using a one-way ANOVA followed by the Tukey–Kramer post hoc test (*$p<0.05$, **$p<0.01$, ns: not significant).

### Effects of PAI-1 deficiency on skeletal muscles with the administration of adenine diets

Grip strength decreased to similar levels in PAI-1$^{+/+}$ and PAI-1$^{-/-}$ mice of both sexes fed adenine diets (Fig 2A). The administration of adenine significantly decreased muscle mass in the whole body to similar levels in male PAI-1$^{+/+}$ and PAI-1$^{-/-}$ mice, whereas PAI-1 deficiency decreased muscle mass in the whole body more in adenine-administered female mice than in

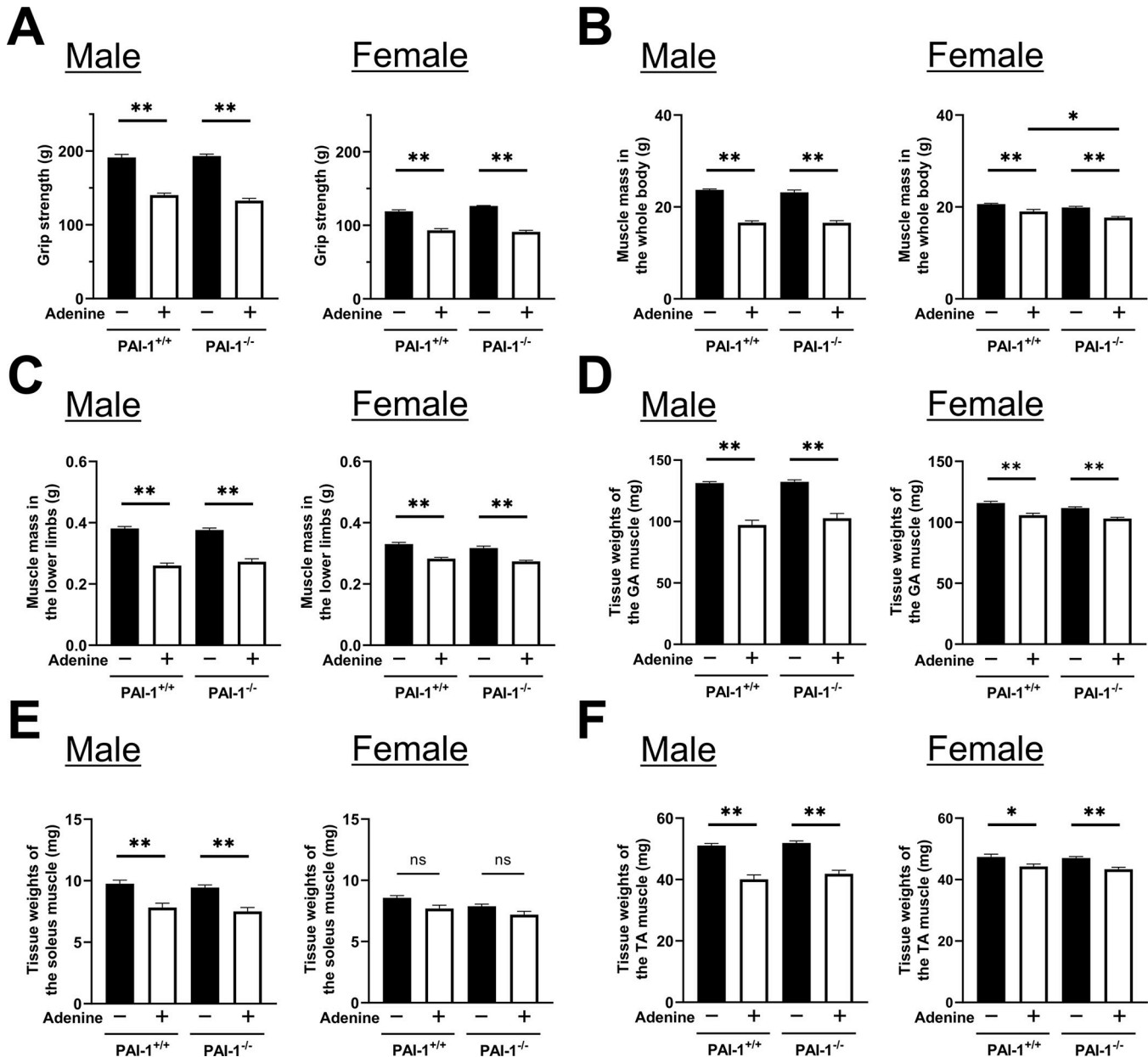

**Fig 2. Effects of PAI-1 deficiency on skeletal muscles with the administration of adenine diets.** (A) Grip strength in PAI-1$^{+/+}$ and PAI-1$^{-/-}$ mice ten weeks after the initiation of adenine administration. (B, C) Muscle mass in the whole body (B) and the lower limbs (C) in PAI-1$^{+/+}$ and PAI-1$^{-/-}$ mice of both sexes analyzed using the qCT system ten weeks after the initiation of adenine administration. (D-F) Tissue weights of the gastrocnemius (GA) (D), soleus (E), and tibialis anterior (TA) muscles (F) in PAI-1$^{+/+}$ and PAI-1$^{-/-}$ mice ten weeks after the initiation of adenine administration. Results are expressed as the means ± SEM of 8 mice per group. Statistical analyses were performed using a one-way ANOVA followed by the Tukey–Kramer post hoc test (*$p<0.05$, **$p<0.01$, ns: not significant).

male mice (Fig 2B). Muscle mass in the lower limbs and the tissue weights of the gastrocnemius, soleus, and tibialis anterior muscles decreased to similar levels in PAI-1$^{+/+}$ and PAI-1$^{-/-}$ mice of both sexes fed the adenine diets; however, the effects of adenine on the tissue weight of the soleus muscle were not significant in female mice (Fig 2C–2F).

## Effects of PAI-1 deficiency on trabecular bone with the administration of adenine diets

In male mice, the administration of adenine significantly decreased trabecular BMD and BV/TV in PAI-1$^{-/-}$ mice, but not in PAI-1$^{+/+}$ mice (Fig 3A and 3B). On the other hand, in female mice, the administration of adenine significantly increased trabecular BMD and BV/TV in adenine-administered PAI-1$^{-/-}$ mice, but not in PAI-1$^{+/+}$ mice. The administration of adenine significantly decreased Tb.N in PAI-1$^{+/+}$ and PAI-1$^{-/-}$ male mice, did not affect Tb.N in female PAI-1$^{+/+}$ mice, and significantly increased Tb.N in female PAI-1$^{-/-}$ mice (Fig 3C). The administration of adenine significantly increased Tb.Th in male PAI-1$^{+/+}$ mice, but not in in PAI-1$^{-/-}$ mice (Fig 3D). In female mice, the administration of adenine did not affect Tb.Th in PAI-1$^{+/+}$ or PAI-1$^{-/-}$ mice. The administration of adenine significantly increased Tb.Sp in PAI-1$^{+/+}$ and PAI-1$^{-/-}$ male mice, but not in female mice (Fig 3E).

## Effects of PAI-1 deficiency on cortical bone with the administration of adenine diets

The administration of adenine decreased CtTMD, Ct.Ar, and Ct.Th in male and female mice, and PAI deficiency did not affect these decreases (Fig 4A–4C). The administration of adenine significantly increased Ct.Po in PAI-1$^{+/+}$ male mice, but not in PAI-1$^{-/-}$ male mice (Fig 4D). On the other hand, the administration of adenine did not affect Ct.Po in PAI-1$^{+/+}$ female mice, but significantly increased Ct.Po in female PAI-1$^{-/-}$ mice. The administration of adenine did not affect Tt.Ar regardless of sex and PAI-1 deficiency (Fig 4E). The administration of adenine significantly decreased Ct.Ar/Tt.Ar regardless of sex and PAI-1 deficiency (Fig 4F).

## Relationships between body weight/body weight change/serum PTH level and muscle/bone parameters

Simple regression analyses with Spearman's rank nonparametric correlation tests were performed on body weight/body weight change/serum PTH level and muscle mass in the whole body, muscle mass in the lower limbs, trabecular BMD, BV/TV, CtTMD, Ct.Ar, or Ct.Th in PAI-1$^{+/+}$ and PAI-1$^{-/-}$ mice ten weeks after the initiation of adenine administration (S1–S3 Tables). Both body weight and body weight change in male mice were significantly and positively corelated to muscle mass in the whole body, muscle mass in the lower limbs, grip strength, trabecular BMD, CtTMD, Ct.Ar, or Ct.Th. This suggests that body weight loss may contribute to a decrease in muscle mass as well as the cortical bone loss in male adenine-induced CKD mice. Body weight in female mice were significantly and positively corelated to muscle mass in the whole body, muscle mass in the lower limbs, grip strength, CtTMD, Ct.Ar, and Ct.Th, although body weight changes in female mice were significantly and positively corelated only to muscle mass in the whole body. Therefore, body weight loss is unlikely to be a primary factor contributing to cortical bone loss in female adenine-induced CKD mice. Serum PTH levels in male mice were significantly corelated to no muscle/bone parameters. On the other hand, serum PTH levels in female mice were significantly and negatively corelated to muscle mass in the whole body, muscle mass in the lower limbs, CtTMD, Ct.Ar and Ct.Th.

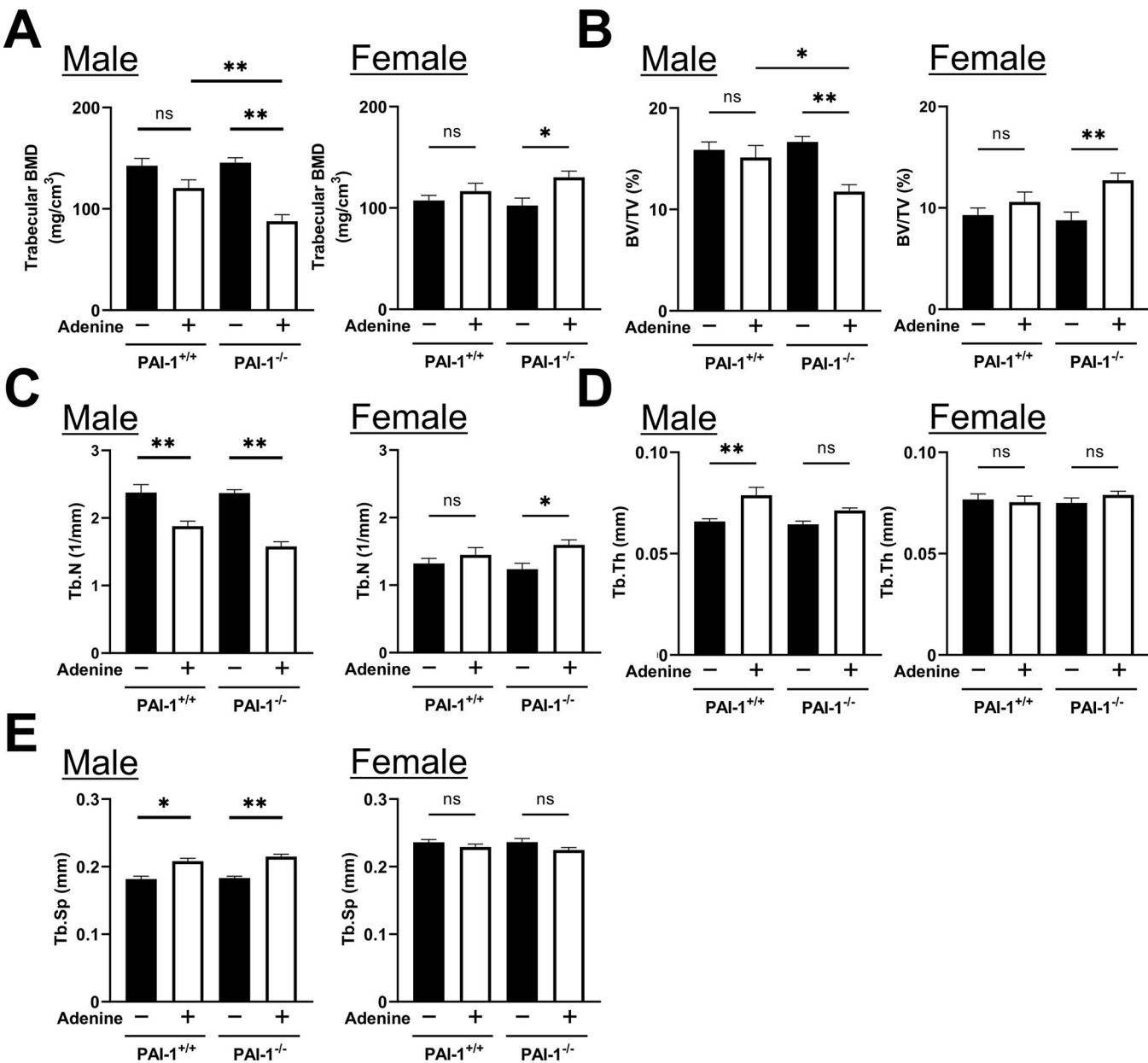

**Fig 3. Effects of PAI-1 deficiency on trabecular bone with the administration of adenine diets.** (A-E) Trabecular bone mineral density (BMD) (A), the bone volume fraction (BV/TV) (B), trabecular number (Tb.N) (C), trabecular thickness (Tb.Th) (D), and trabecular separation (Tb.Sp) (E) in PAI-1$^{+/+}$ and PAI-1$^{-/-}$ mice of both sexes analyzed using the μCT system ten weeks after the initiation of adenine administration. Results are expressed as the means ± SEM of 8 mice per group. Statistical analyses were performed using a one-way ANOVA followed by the Tukey–Kramer post hoc test (*$p<0.05$, **$p<0.01$, ns: not significant).

## Discussion

In the present study, bone loss was observed in the cortical bone parameters (CtTMD, Ct.Ar, and Ct.Th) of adenine-administered wild-type mice, but not in trabecular bone parameters (BMD and BV/TV). These results are consistent with our previous findings from 5/6 nephrectomized CKD model mice [5]. However, PAI-1 deficiency did not affect CKD-induced cortical bone loss and muscle wasting or decreases in grip strength in male and female mice in the present study, indicating that PAI-1 is not crucial for cortical bone loss and sarcopenia induced by CKD in mice. Since we previously suggested the involvement of PAI-1 in

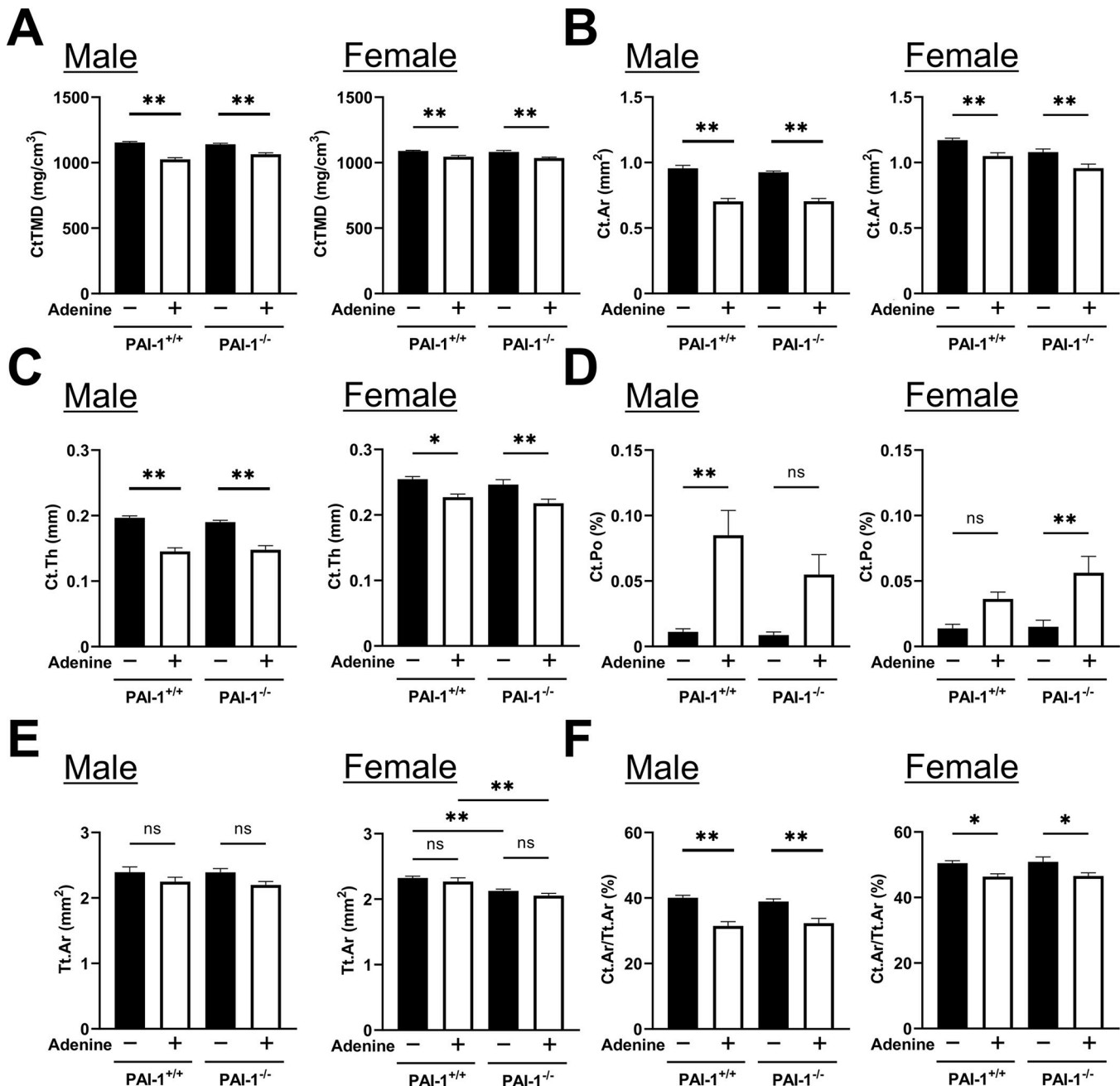

**Fig 4. Effects of PAI-1 deficiency on cortical bone with the administration of adenine diets.** (A-F) Cortical tissue mineral density (CtTMD) (A), cortical bone area (Ct.Ar) (B), average cortical thickness (Ct.Th) (C), cortical porosity (Ct.Po) (D), total cross-sectional area (Tt.Ar) (E), and the cortical area fraction (Ct.Ar/Tt.Ar) (F) in PAI-1$^{+/+}$ and PAI-1$^{-/-}$ mice analyzed using the μCT system ten weeks after the initiation of adenine administration. Results are expressed as the means ± SEM of 8 mice per group. Statistical analyses were performed using a one-way ANOVA followed by the Tukey–Kramer post hoc test (*$p < 0.05$, **$p < 0.01$, ns: not significant).

trabecular bone loss and delayed bone repair induced by a diabetic state and glucocorticoid excess in mice [8–11], and a diabetic state and glucocorticoid excess induce a chronic inflammation state and excessive oxidative stress, the pathogenesis of CKD-MBD may markedly differ from the pathophysiology of bone loss induced by diabetes and glucocorticoid excess. Otherwise, PAI-1 is not crucial for cortical bone metabolism and loss as well as sarcopenia.

Hyperparathyroidism is crucial for the pathophysiology of CKD-MBD and is caused by impaired calcium reabsorption and vitamin D deficiency. Hyperparathyroidism decreases bone mass more prominently in cortical bone than in trabecular bone [21]. However, the effects of adenine administration on serum PTH levels were not significant in the present study, although adenine administration seemed to elevate serum PTH levels without any significant differences in female mice. It is puzzling that the increases in serum PTH level seemed very slight despite the bone phenotypes characteristic of CKD in the present study. Previous studies have shown an increase in serum PTH levels following adenine administration [22–24]. In the present study, CKD-induced mice were initially fed a high-dose adenine-containing diet for 2 weeks to induce renal dysfunction, followed by a low-dose adenine-containing diet for 8 weeks to sustain kidney damage. Although serum PTH levels in CKD-induced mice might have been elevated immediately after the initiation of adenine administration, their renal dysfunction might have partly recovered sufficiently to maintain calcium metabolism without significant elevated PTH levels. The negative correlation of serum PTH levels and cortical bone parameter in female mice possibly supports that cortical bone loss were caused by the increase of serum PTH levels. Moreover, bone loss has been reported in thyroparathyroidectomy- and 5/6 nephrectomy-treated mice, suggesting that CKD can induce bone loss independent of hyperparathyroidism [25]. Taken into account with that BUN and serum creatinine levels were elevated in adenine-administrated mice, PTH-independent mechanisms might contribute to the bone loss in mice in the present study.

We previously reported sex differences in the role of PAI-1 in bone remodeling in pathogenetic mice [11,13]. PAI-1 deficiency rescued diabetes-induced trabecular bone loss as well as the bone expressions of Runx2, Osterix, and ALP in female mice, but not male [11]. Moreover, the addition of active PAI-1 inhibited ALP activity and mineralization in osteoblasts derived from female mice, but not male [11], and PAI-1 deficiency decreased and increased the mRNA expressions of osteopontin and matrix gla protein in osteoblasts derived from female mice, but not male, respectively [13]. On the other hand, PAI-1 deficiency blunted bone loss induced by glucocorticoid in both male and female mice without sex differences [8]. Endogenous PAI-1 supports the early-stage osteogenic differentiation of mesenchymal stem cells derived from mice of both sexes [12]. Taken together, the involvement of sex differences in PAI-1 effects on bone metabolism might be different due to osteoblast differentiation stage or hormonal effects on bone. In the present study, PAI-1 deficiency significantly decreased BMD and BV/TV in the trabecular bone of male mice with adenine administration, although it significantly increased BMD and BV/TV in the trabecular bone of female mice with adenine administration. Taken together our data and previous evidence, endogenous PAI-1 effects on the osteogenesis of mesenchymal stem cells might be predominant in the absence of bone catabolic effects of exogenous PAI-1 on osteogenic differentiation and mineralization in male CKD mice. In female CKD mice, exogenous PAI-1 effects on osteogenic differentiation and mineralization might be predominant, compared to bone protective effects of endogenous PAI-1.

In the present study, the changes in trabecular bone parameters, such as Tb.N, Tb.Th, and Tb.Sp, were observed in only male PAI-1$^{+/+}$ mice, but not in female. Previous studies on adenine-induced CKD rodent models have shown sex differences [26], in which adenine administration decreased and increased plasma testosterone and estrogen levels only in male mice, but not female, respectively. The sex differences of sex hormone levels by adenine administration might contribute to the different effects on trabecular bone parameters between male and female mice. Differences in the effects of PAI-1 deficiency on serum calcium and phosphorus levels may also modulate its effects on trabecular bone parameters partly by affecting bone remodeling and mineralization. In the present study, serum calcium levels were significantly

increased in only PAI-1$^{+/+}$ male mice, but not female. Its reasons have remained unclear. In the present study, the severity of CKD induced by adenine administration based on serum BUN and creatinine levels seemed predominant in male mice than in female. Sex differences in adenine-induced CKD rodent model were reported previously [26], in which adenine-fed females had less decline in kidney function than adenine-fed males. Sex differences of the severity of renal dysfunction might partly influence the changes in serum calcium levels due to the administration of adenine. Furthermore, the roles of PAI-1 in bone remodeling may depend on the stage of CKD.

The adenine-induced CKD rodent model is one of the most frequently used models for investigating the pathophysiology of CKD in mice [19]. This model is superior to other CKD models due to the lack of surgical interventions, its low mortality rate, and the consistent induction of a stable pathology. However, a limitation of the adenine-induced CKD model is a decreased food intake due to the distinct smell and taste of adenine. In the present study, the administration of adenine reduced food intake and body weight in both PAI-1$^{+/+}$ and PAI-1$^{-/-}$ mice. Additionally, CKD induced by the administration of adenine based on serum BUN and creatinine levels was more severe in male mice than in female mice. Therefore, malnutrition or the severity of CKD due to sex differences may have affected muscle wasting and bone disorders in our CKD model mice.

Serum calcium levels typically decrease as CKD progresses in humans due to a decrease in the renal activation of vitamin D and subsequent vitamin D deficiency. However, secondary hyperparathyroidism associated with disturbances in calcium and vitamin D metabolism induced by CKD corrects serum calcium levels to within normal ranges when CKD is not severe [27]. In the present study, serum calcium levels significantly increased in adenine-induced CKD mice, except for female PAI-1$^{+/+}$ mice. Previous studies reported increases or no changes in serum calcium levels in adenine-induced CKD rodent models [28,29]. A decrease in serum calcium levels stimulates the secretion and synthesis of PTH as well as para-thyroid gland hyperplasia. Parathyroid gland hyperplasia gradually progresses in humans, typically over many months or even years. However, previous rodent studies showed that parathyroid gland hyperplasia may occur within a few days or weeks [27,30]. Additionally, mice were initially fed a high-dose adenine-containing diet, which was later switched to a low-dose adenine-containing diet to induce the pathogenesis of CKD through the administration of adenine in the present study. Therefore, mice may initially have developed severe CKD with parathyroid gland hyperplasia and then recovered to milder CKD with enlarged parathyroid glands. In the present study, a significant elevation in serum PTH levels were not observed in mice treated with adenine in the presence or absence of PAI-1 deficiency. Therefore, the impact of increased serum calcium on bone and mineral metabolism might be minor in CKD-induced bone changes in mice, although why serum calcium levels in the present study were elevated by adenine administration have remained unknown. The drug effects of adenine other than renal dysfunction, the differences of mouse strains, or the balance of diet and renal dysfunction might affect serum calcium levels.

In CKD patients, serum phosphorus levels are elevated due to the impaired renal excretion of phosphorus, which is partly related to Klotho inactivity or elevated FGF23 levels [31]. In the present study, PAI-1 deficiency significantly increased serum phosphorus levels in the control group and mitigated the elevated serum phosphorus level induced by the administration of adenine in female mice, but not male mice. FGF23, a peptide hormone secreted by osteocytes and osteoblasts, inhibits the reabsorption of proximal tubular phosphate by interacting with Klotho and also suppresses intestinal phosphate absorption by decreasing renal vitamin D activation [32]; however, elevated serum phosphorus levels have been reported in FGF23-deficient mice [33]. Moreover, PAI-1 has been shown to inhibit the plasminogen activator-dependent

cleavage of FGF23 [34]. Therefore, PAI-1 deficiency may have promoted the reabsorption of phosphorus by accelerating the cleavage of FGF23, which may explain the higher serum phosphorus levels in PAI-1$^{-/-}$ female mice than in PAI-1$^{+/+}$ female mice. Based on the potential down-regulation of the FGF23-Klotho axis prior to the induction of CKD due to decreased FGF23 levels, PAI-1 deficiency may not have elevated serum phosphorus levels following the induction of CKD. Nevertheless, the changes induced in serum phosphorus levels by the pathogenesis of CKD were consistent between PAI-1$^{+/+}$ and PAI-1$^{-/-}$ male mice. Since we previously reported sex differences in PAI-1 activity across various pathophysiological states [11,13,14], sex differences may exist in the roles of PAI-1 in regulating serum phosphorus levels.

## Conclusion

The present study showed that PAI-1 deficiency did not affect muscle wasting or cortical bone loss in adenine-induced CKD model mice. These results suggest that PAI-1 is not critical for the pathophysiology of CKD-MBD or CKD-induced sarcopenia in mice. However, PAI-1 may be partly related to bone metabolism in trabecular bone in the CKD state with sex differences. Further studies are needed to clarify the cellular and molecular mechanisms by which PAI-1 affects bone remodeling in the CKD state.

## Supporting information

**S1 Fig. The genotyping data of PAI-1+/+ and PAI-1-/- mice.** (Left) Genotyping by PCR was performed to identify PAI-1$^{-/-}$ mice with Pai-1 primer. The absence of a PCR product indicates the genomic knockout of Pai-1. The band shows a PCR product of 1340 bp. (Right) Genotyping by PCR was performed to confirm PAI-1$^{-/-}$ mice with Neor primer. The presence of a PCR product indicates the genomic knock in of Neor in the process of gene editing. The band shows a PCR product of 450 bp. The lower bands represent unspecific primer dimers.
(TIF)

**S2 Fig. Simple regression analyses between serum PTH levels and serum calcium /phosphorus levels.** 2D scatter plots with regression line of the relationship between serum PTH levels and serum calcium (upper) and phosphorus (lower) levels in all males (left) and females (right) mice used in the present study. A simple regression analysis was performed using Spearman's rank nonparametric correlation test. ($r$: Spearman's rank correlation coefficient; $p$: p value).
(TIF)

**S1 Table. Simple regression analyses between body weight and muscle/bone parameters.** Simple regression analyses were performed between body weight and muscle mass in the whole body, muscle mass in the lower limbs, trabecular bone mineral density (BMD), the bone volume fraction (BV/TV), cortical tissue mineral density (CtTMD), cortical bone area (Ct.Ar), cortical thickness (Ct.Th) in PAI-1$^{+/+}$ and PAI-1$^{-/-}$ mice with or without adenine administration (n = 8 mice in each group). A simple regression analysis was performed with Spearman's rank nonparametric correlation test. (r: Spearman's rank correlation coefficient, **$p$<0.01).
(DOCX)

**S2 Table. Simple regression analyses between body weight change and muscle/bone parameters.** Simple regression analyses were performed between body weight change and muscle mass in the whole body, muscle mass in the lower limbs, trabecular bone mineral density (BMD), the bone volume fraction (BV/TV), cortical tissue mineral density (CtTMD), cortical bone area (Ct.Ar), cortical thickness (Ct.Th) in female PAI-1$^{+/+}$ and PAI-1$^{-/-}$ mice with or

without adenine administration (n = 8 mice in each group). A simple regression analysis was performed with Spearman's rank nonparametric correlation test. (r: Spearman's rank correlation coefficient, $^*p<0.05$, $^{**}p<0.01$).
(DOCX)

**S3 Table. Simple regression analyses between serum PTH levels and muscle/bone parameters.** Simple regression analyses were performed between serum PTH levels and muscle mass in the whole body, muscle mass in the lower limbs, trabecular bone mineral density (BMD), the bone volume fraction (BV/TV), cortical tissue mineral density (CtTMD), cortical bone area (Ct.Ar), cortical thickness (Ct.Th) in male or female PAI-1$^{+/+}$ and PAI-1$^{-/-}$ mice with or without adenine administration (n = 8 mice in each group). A simple regression analysis was performed with Spearman's rank nonparametric correlation test. (r: Spearman's rank correlation coefficient, $^*p<0.05$, $^{**}p<0.01$).
(DOCX)

**S1 File. A summary of all values used to create the graphs.**
(XLSX)

## Author Contributions

**Conceptualization:** Hiroshi Kaji.

**Data curation:** Yuya Mizukami.

**Formal analysis:** Yuya Mizukami.

**Funding acquisition:** Hiroshi Kaji.

**Investigation:** Yuya Mizukami, Naoyuki Kawao, Takashi Ohira, Hisatoshi Yamao.

**Methodology:** Yuya Mizukami, Hiroshi Kaji.

**Project administration:** Hiroshi Kaji.

**Supervision:** Hiroshi Kaji.

**Validation:** Naoyuki Kawao, Takashi Ohira.

**Visualization:** Yuya Mizukami.

**Writing – original draft:** Yuya Mizukami.

**Writing – review & editing:** Naoyuki Kawao, Takashi Ohira, Kiyotaka Okada, Osamu Matsuo, Hiroshi Kaji.

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
