## [Decision Letter · Decision Letter 0]

29 Jul 2024

PONE-D-24-27375Effects of plasminogen activator inhibitor-1 deficiency on bone disorders and sarcopenia caused by adenine-induced renal dysfunction in micePLOS ONE

Dear Dr. Kaji,

Thank you for submitting your manuscript to PLOS ONE. After careful consideration, we feel that it has merit but does not fully meet PLOS ONE’s publication criteria as it currently stands. Therefore, we invite you to submit a revised version of the manuscript that addresses the points raised during the review process.

We look forward to receiving your revised manuscript.

Kind regards,

Toshio Matsumoto

Academic Editor

PLOS ONE

Journal Requirements:

2. To comply with PLOS ONE submissions requirements, in your Methods section, please provide additional information regarding the experiments involving animals and ensure you have included details on methods of sacrifice.

Reviewers' comments:

Reviewer's Responses to Questions

**Comments to the Author**

1. Is the manuscript technically sound, and do the data support the conclusions?

Reviewer #1: Partly

Reviewer #2: Yes

2. Has the statistical analysis been performed appropriately and rigorously? 

Reviewer #1: No

Reviewer #2: Yes

3. Have the authors made all data underlying the findings in their manuscript fully available?

Reviewer #1: Yes

Reviewer #2: Yes

4. Is the manuscript presented in an intelligible fashion and written in standard English?

Reviewer #1: Yes

Reviewer #2: Yes

5. Review Comments to the Author

Reviewer #1: General comments

In this manuscript, the authors analyzed effect of PAI-1 gene deletion on bone and muscle metabolism in an adenine-induced CKD rat model. Substantial BW loss observed in this model seems to be a significant problem, and this reviewer is not convinced that bone and muscle loss is indeed caused by renal failure. Besides, “sex differences” in PAI-1 effect may be somewhat inconsistent between experiments (including those already published), and it is hard to understand what PAI-1 is doing in each sex.

Specific comments

1) There is a significant concern about variability in the genetic background. Please explain “a mixed C57BL/6J (81.25%) and 129/SvJ (18.75%) background” and how the mice were prepared. Discuss potential experimental problems caused by such genetic heterogeneity.

2) BW in Figure 1B and muscle mass in Figure 2B show the same pattern, suggesting changes in muscle mass was caused by BW loss rather than specific effect of CKD on muscle. It would be useful to examine correlations between BW and muscle mass (or changes in the two parameters) in all the mice to see if sarcopenia can be mostly explained by BW loss or not.

3) The authors should also examine correlations between changes in BW and bone parameters.

4) The authors should measure PTH levels in the serum and look at the correlation between PTH and Ca. This is a must. Drawing a 2D scatter plot would be informative. It is critically important to confirm that PTH levels were indeed elevated in the presence of hypercalcemia and hyperphosphatemia. And I wonder if hypercalcemia occurred in all the animals.

5) In Figure 1G, the serum P levels were higher in PAI-1-/- compared to PAI-1 +/+ without treatment. Significantly elevated serum P at the basal level is surprising and may profoundly affect the whole results. If this is true, all the previous experiments by the authors using PAI-1 KO would also have to be totally reassessed from a different view.

How many times in independent experiments did the authors confirm increased P in these mice? It would also be necessary to examine P levels at different ages.

Demonstration of low FGF23 only in female rats in the presence of high P would be convincing.

6) In Figure 1F, serum calcium increased in PAI-1+/+ only in males, not in females. Why?

7) Similarly, in Figures 3C-E, changes in trabecular parameters were only significant in males. Are there sex differences in response to adenine in wild-type animals?

8) In Figure 3A, trabecular BMD in PAI-1-/- males decreased by treatment with adenine whereas it increased in PAI-1-/- females. The authors often describe “sex differences” in PAI-1 action, but the authors should provide some reasonable explanation based on all the previous experiments the authors already reported. After all, what is the role of PAI-1 in bone in males and females? At least, the authors should discuss the current results together with all the previous experiments and provide a consistent explanation. Just describing the results as “sex differences” is not enough. Otherwise, it is just confusing, and there would be no advances in our understanding of the function of PAI-1 in bone.

Reviewer #2: GENERAL COMMENTS

The authors examined the effects of PAI-1 deficiency on bone disorders and sarcopenia on CKD model mice. The manuscript is well-written and easy to follow.

Some, but not all, of the points that should be addressed are listed below.

SPECIFIC COMMENTS

#1 Please indicate the genotyping data in PAI-1 KO mice.

#2 The reviewer wonder if the total muscle mass was too heavy considering the mice body weight indicated in Fig 2B.

#3 Please show the method for measuring skeletal muscle weight on Fig2 B-F.

#4 If possible, please show the data of PTH and/or ALP in mice.

6. PLOS authors have the option to publish the peer review history of their article (what does this mean?). If published, this will include your full peer review and any attached files.

Reviewer #1: No

Reviewer #2: No

---

## [Author Response · Author response to Decision Letter 0]

12 Sep 2024

All of our responses to the reviewers' comments, including graphical data, are provided in the "Response to Reviewers" file. The responses, excluding additional figures, tables, and reference information, are provided below.

Responses to the comments of Reviewer #1

Re: “General comments”

“In this manuscript, the authors analyzed effect of PAI-1 gene deletion on bone and muscle metabolism in an adenine-induced CKD rat model. Substantial BW loss observed in this model seems to be a significant problem, and this reviewer is not convinced that bone and muscle loss is indeed caused by renal failure. Besides, “sex differences” in PAI-1 effect may be somewhat inconsistent between experiments (including those already published), and it is hard to understand what PAI-1 is doing in each sex.”

(Response)

We would like to express our gratitude to the reviewer for their important insights on our study. 

Re: “Specific comments”

“1) There is a significant concern about variability in the genetic background. Please explain “a mixed C57BL/6J (81.25%) and 129/SvJ (18.75%) background” and how the mice were prepared. Discuss potential experimental problems caused by such genetic heterogeneity.”

(Response)

PAI-1+/+ and PAI-1−/− mice with a mixed C57BL/6J (81.25%) and 129/SvJ (18.75%) background were originally generated by Professor D. Collen at the University of Leuven, Belgium. The methods for generating these PAI-1+/+ and PAI-1−/− mice, along with the genotyping data, have been detailed in a previous study (Carmeliet P et al., J Clin Invest. 1993;92(6):2746-55). In 1996, Professor D. Collen provided the initial PAI-1+/+ and PAI-1−/− mice, which were subsequently bred in the animal facility at Kindai University. To minimize the effects of the mixed mouse strain, we obtained male and female mice with heterozygous PAI-1 (PAI-1+/-) gene deficiency by crossbreeding PAI-1+/+ and PAI-1−/− mice. These heterozygous littermates were then repeatedly bred. For the present study, PAI-1+/+ and PAI-1−/− mice were prepared by breeding homozygous littermates obtained from heterozygous breeding. The genotypes were determined by PCR analysis (S Fig.1). Consequently, the genetic background of the PAI-1+/+ and PAI-1−/− mice used in this study is nearly identical. These sentences about the generation and the preparation of PAI-1+/+ and PAI-1−/− mice were added to Materials and Methods (page 6, line 83-page 93), and the following references [20] were added to References.

Re: “2) BW in Figure 1B and muscle mass in Figure 2B show the same pattern, suggesting changes in muscle mass was caused by BW loss rather than specific effect of CKD on muscle. It would be useful to examine correlations between BW and muscle mass (or changes in the two parameters) in all the mice to see if sarcopenia can be mostly explained by BW loss or not.”

“3) The authors should also examine correlations between changes in BW and bone parameters.”

(Response)

Simple regression analyses with Spearman’s rank nonparametric correlation tests were performed on body weight/body weight change and muscle mass in the whole body, muscle mass in the lower limbs, grip strength, trabecular BMD, BV/TV, CtTMD, Ct.Ar, or Ct.Th in PAI-1+/+ and PAI-1-/- mice ten weeks after the initiation of adenine administration (S1,2 Table). Both body weight and body weight change in male mice were significantly and positively corelated to muscle mass in the whole body, muscle mass in the lower limbs, grip strength, trabecular BMD, CtTMD, Ct.Ar, or Ct.Th. This suggests that body weight loss may contribute to a decrease in muscle mass as well as the cortical bone loss in male adenine-induced CKD mice. Body weight in female mice were significantly and positively corelated to muscle mass in the whole body, muscle mass in the lower limbs, grip strength, CtTMD, Ct.Ar, and Ct.Th, although body weight changes in female mice were significantly and positively corelated only to muscle mass in the whole body. Therefore, body weight loss is unlikely to be a primary factor contributing to cortical bone loss in female adenine-induced CKD mice. These comments were added to Results (page 15, line 264-279), and the data about the simple regression analyses were added as S1,2 Table.

Re: “4) The authors should measure PTH levels in the serum and look at the correlation between PTH and Ca. This is a must. Drawing a 2D scatter plot would be informative. It is critically important to confirm that PTH levels were indeed elevated in the presence of hypercalcemia and hyperphosphatemia. And I wonder if hypercalcemia occurred in all the animals.”

(Response)

We performed the additional experiments to measure serum PTH levels by enzyme-linked immunosorbent assay (ELISA). The effects of adenine administration on serum PTH levels were not significant in all groups, although adenine administration tended to elevate serum PTH levels without any significant differences in female mice and PAI-1 deficiency significantly increased serum PTH levels in control male mice (Fig 1H). These comments were added to Results (page 11, line 181-185) and the data were added as new Figure 1H. Moreover, simple regression analyses with Spearman’s rank nonparametric correlation tests were performed on serum PTH level and serum calcium and phosphorus level. Serum PTH levels were significantly and positively corelated to serum calcium and phosphorus levels in only female mice, but not male, in the simple regression analyses. These comments were added to Results (page 12, line 185-187) and the relationships between serum PTH level and serum calcium, and phosphorus levels (a 2D scatter plot) were shown as S2 Figure. Furthermore, simple regression analyses with Spearman’s rank nonparametric correlation tests were performed on serum PTH level and muscle mass in the whole body, muscle mass in the lower limbs, grip strength, trabecular BMD, BV/TV, CtTMD, Ct.Ar and Ct.Th, in PAI-1+/+ and PAI-1-/- mice ten weeks after the initiation of adenine administration. Serum PTH levels in male mice were significantly corelated to no muscle/bone parameters. On the other hand, serum PTH levels in female mice were significantly and negatively corelated to muscle mass in the whole body, muscle mass in the lower limbs, CtTMD, Ct.Ar and Ct.Th. These comments were added to Results (page 16, line 279-282) and the data were added as new S3 Table. 

Hyperparathyroidism is crucial for the pathophysiology of CKD-MBD and is caused by impaired calcium reabsorption and vitamin D deficiency. Hyperparathyroidism decreases bone mass more prominently in cortical bone than in trabecular bone [21]. However, the effects of adenine administration on serum PTH levels were not significant in the present study, although adenine administration seemed to elevate serum PTH levels without any significant differences in female mice. It is puzzling that the increases in serum PTH level seemed very slight despite the bone phenotypes characteristic of CKD in the present study. Previous studies have shown an increase in serum PTH levels following adenine administration (Metzger CE, et al., Calcif Tissue Int. 2020;106(4):392-400; Metzger CE, et al., Bone. 2021 Jul;148:115963; De Maré A, et al., J Bone Miner Res. 2022 Apr;37(4):687-699.). In the present study, CKD-induced mice were initially fed a high-dose adenine-containing diet for 2 weeks to induce renal dysfunction, followed by a low-dose adenine-containing diet for 8 weeks to sustain kidney damage. Although serum PTH levels in CKD-induced mice might have been elevated immediately after the initiation of adenine administration, their renal dysfunction might have partly recovered sufficiently to maintain calcium metabolism without significant elevated PTH levels. The negative correlation of serum PTH levels and cortical bone parameter in female mice possibly supports that cortical bone loss were caused by the increase of serum PTH levels. Moreover, bone loss has been reported in thyroparathyroidectomy- and 5/6 nephrectomy-treated mice, suggesting that CKD can induce bone loss independent of hyperparathyroidism (Iwasaki Y, et al., Bone. 2015;81:247-254). Taken into account with that BUN and serum creatinine levels were elevated in adenine-administrated mice, PTH-independent mechanisms might contribute to the bone loss in mice in the present study. These comments were added to Discussion (page 17, line 298-318). The references [22-25] were added to References.

In the present study, a significant elevation in serum PTH levels were not observed in mice treated with adenine in the presence or absence of PAI-1 deficiency. Therefore, the impact of increased serum calcium on bone and mineral metabolism might be minor in CKD-induced bone changes in mice, although why serum calcium levels in the present study were elevated by adenine administration have remained unknown. The drug effects of adenine other than renal dysfunction, the differences of mouse strains, or the balance of diet and renal dysfunction might affect serum calcium levels. These comments were added to Discussion (page 22, line 384-390).

Re: “5) In Figure 1G, the serum P levels were higher in PAI-1-/- compared to PAI-1+/+ without treatment. Significantly elevated serum P at the basal level is surprising and may profoundly affect the whole results. If this is true, all the previous experiments by the authors using PAI-1 KO would also have to be totally reassessed from a different view. How many times in independent experiments did the authors confirm increased P in these mice? It would also be necessary to examine P levels at different ages. Demonstration of low FGF23 only in female rats in the presence of high P would be convincing”

(Response)

Unfortunately, we have not measured serum phosphorus levels in PAI-1+/+ and PAI-1-/- mice with or without adenine administration several times in the present study. Moreover, we have not measured serum FGF23 levels for the limited amount of blood samples from mice. However, we agree that phosphorus metabolism might be partly related to sex differences of PAI-1 effects on bone in mice, since there is a previous report about the relationship between PAI-1 and FGF23 [34], as we described in Discussion. We would like to plan the further studies to clarify evaluate the role of PAI-1 in the phosphorus metabolism in the future. 

Re: “6) In Figure 1F, serum calcium increased in PAI-1+/+ only in males, not in females. Why?”

(Response)

In the present study, serum calcium levels were significantly increased in only PAI-1+/+ male mice, but not female. Its reasons have remained unclear. In the present study, the severity of CKD induced by adenine administration based on serum BUN and creatinine levels seemed predominant in male mice than in female. Sex differences in adenine-induced CKD rodents model were reported previously (Diwan V et al., Am J Physiol Renal Physiol. 2014;307(11):F1169-78.), in which adenine-fed females had less decline in kidney function than adenine-fed males. Sex differences of the severity of renal dysfunction might partly influence the changes in serum calcium levels due to the administration of adenine. These comments were added to Discussion (page 20, line 348-357), and a reference [26] was added to References.

Re: “7) Similarly, in Figures 3C-E, changes in trabecular parameters were only significant in males. Are there sex differences in response to adenine in wild-type animals?”

(Response)

In the present study, the changes in trabecular bone parameters, such as Tb.N, Tb.Th, and Tb.Sp, were observed in only male PAI-1+/+ mice, but not in female. Previous studies on adenine-induced CKD rodent models have shown sex differences (Diwan V et al., Am J Physiol Renal Physiol. 2014;307(11):F1169-78.), in which adenine administration decreased and increased plasma testosterone and estrogen levels only in male mice, but not female, respectively. The sex differences of sex hormone levels by adenine administration might contribute to the different effects on trabecular bone parameters between male and female mice. These comments were added to Discussion (page 19, line 340-346).

Re: “8) In Figure 3A, trabecular BMD in PAI-1-/- males decreased by treatment with adenine whereas it increased in PAI-1-/- females. The authors often describe “sex differences” in PAI-1 action, but the authors should provide some reasonable explanation based on all the previous experiments the authors already reported. After all, what is the role of PAI-1 in bone in males and females? At least, the authors should discuss the current results together with all the previous experiments and provide a consistent explanation. Just describing the results as “sex differences” is not enough. Otherwise, it is just confusing, and there would be no advances in our understanding of the function of PAI-1 in bone.”

(Response)

We revised the explanation regarding the sex differences in PAI-1 effects on bone remodeling (Discussion, page 18, line 319-339), as follows: 

“We previously reported sex differences in the role of PAI-1 in bone remodeling in pathogenetic mice [11,13]. PAI-1 deficiency rescued diabetes-induced trabecular bone loss as well as the bone expressions of Runx2, Osterix, and ALP in female mice, but not male [11]. Moreover, the addition of active PAI-1 inhibited ALP activity and mineralization in osteoblasts derived from female mice, but not male [11], and PAI-1 deficiency decreased and increased the mRNA expressions of osteopontin and matrix gla protein in osteoblasts derived from female mice, but not male, respectively [13]. On the other hand, PAI-1 deficiency blunted bone loss induced by glucocorticoid in both male and female mice without sex differences [8]. Endogenous PAI-1 supports the early-stage osteogenic differentiation of mesenchymal stem cells derived from mice of both sexes [12]. Taken together, the involvement of sex differences in PAI-1 effects on bone metabolism might be different due to osteoblast differentiation stage or hormonal effects on bone. In the present study, PAI-1 deficiency significantly decreased BMD and BV/TV in the trabecular bone of male mice with adenine administration, although it significantly increased BMD and BV/TV in the trabecular bone of female mice with adenine administration. Taken together our data and previous evidence, endogenous PAI-1 effects on the osteogenesis of mesenchymal stem cells might be predominant in the absence of bone catabolic effects of exogenous PAI-1 on osteogenic differentiation and mineralization in male CKD mice. In female CKD mice, exogenous PAI-1 effects on osteogenic differentiation and mineralization might be predominant, compared to bone protective effects of endogenous PAI-1.”

Responses to the comments of Reviewer #2

Re: “General comments” 

“The authors examined the effects of PAI-1 deficiency on bone disorders and sarcopenia on CKD model mice. The manuscript is well-written and easy to follow. Some, but not all, of the points that should be addressed are listed below.”

(Response)

We would like to express our gratitude for the positive comments from the reviewer on our study.

Re: “SPECIFIC COMMENTS” 

“#1 Please indicate the genotyping data in PAI-1 KO mice.”

(Response)

The genotyping data were added as supplementary Figure 1, as follows. Also, the methods about genotyping were added to Materials and Methods (page 6, line 89-91).

Re: “#2 The reviewer wonder if the total muscle mass was too heavy considering the mice body weight indicated in Fig 2B.”

(Response)

Total muscle mass measured as “lean body mass” by qCT system seemed to be much higher in mice, compared to humans, which were reported in numerous previous papers (Iemura S, Kawao N, Okumoto K, Akagi M, Kaji H. Role of irisin in androgen-deficient muscle wasting and osteopenia in mice. J Bone Miner Metab. 2020 Mar;38(2):161-171.; Kawao N, Iemura S, Kawaguchi M, Mizukami Y, Takafuji Y, Kaji H. Role of irisin in effects of chronic exercise on muscle and bone in ovariectomized mice. J Bone Miner Metab. 2021 Jul;39(4):547-557.; Kawao N, Kawaguchi M, Ohira T, Ehara H, Mizukami Y, Takafuji Y, Kaji H. Renal failure suppresses muscle irisin expression, and irisin blunts cortical bone loss in mice. J Cachexia Sarcopenia Muscle. 2022 Feb;13(1):758-

---

## [Decision Letter · Decision Letter 1]

27 Sep 2024

Effects of plasminogen activator inhibitor-1 deficiency on bone disorders and sarcopenia caused by adenine-induced renal dysfunction in mice

PONE-D-24-27375R1

Dear Dr. Kaji,

We’re pleased to inform you that your manuscript has been judged scientifically suitable for publication and will be formally accepted for publication once it meets all outstanding technical requirements.

Kind regards,

Toshio Matsumoto

Academic Editor

PLOS ONE

Additional Editor Comments (optional):

Reviewers' comments:

Reviewer's Responses to Questions

**Comments to the Author**

1. If the authors have adequately addressed your comments raised in a previous round of review and you feel that this manuscript is now acceptable for publication, you may indicate that here to bypass the “Comments to the Author” section, enter your conflict of interest statement in the “Confidential to Editor” section, and submit your "Accept" recommendation.

Reviewer #1: All comments have been addressed

Reviewer #2: All comments have been addressed

2. Is the manuscript technically sound, and do the data support the conclusions?

Reviewer #1: (No Response)

Reviewer #2: Yes

3. Has the statistical analysis been performed appropriately and rigorously? 

Reviewer #1: (No Response)

Reviewer #2: Yes

4. Have the authors made all data underlying the findings in their manuscript fully available?

Reviewer #1: (No Response)

Reviewer #2: Yes

5. Is the manuscript presented in an intelligible fashion and written in standard English?

Reviewer #1: (No Response)

Reviewer #2: Yes

6. Review Comments to the Author

Reviewer #1: (No Response)

Reviewer #2: The authors have answered all of my question. The manuscript was revised well and is suitable for publication in PLOS ONE.

7. PLOS authors have the option to publish the peer review history of their article (what does this mean?). If published, this will include your full peer review and any attached files.

Reviewer #1: No

Reviewer #2: **Yes: **Itsuro Endo

---

## [Editor Report · Acceptance letter]

2 Oct 2024

PONE-D-24-27375R1 

PLOS ONE

Dear Dr. Kaji, 

I'm pleased to inform you that your manuscript has been deemed suitable for publication in PLOS ONE. Congratulations! Your manuscript is now being handed over to our production team.

Kind regards, 

on behalf of

Dr. Toshio Matsumoto 

Academic Editor

PLOS ONE